# Inherent Limits on Topology-Based Link Prediction

**Justus Isaiah Hibshman**                                                                          *jhibshma@nd.edu*
*Department of Computer Science and Engineering*
*University of Notre Dame*

**Tim Weninger**                                                                                    *tweninger@nd.edu*
*Department of Computer Science and Engineering*
*University of Notre Dame*

**Reviewed on OpenReview:** *https://openreview.net/forum?id=izL3B8dPx1*

## Abstract

Link prediction systems (*e.g.* recommender systems) typically use graph topology as one of their main sources of information. However, automorphic symmetries in graphs limit how informative topology is for link prediction. We calculate hard upper bounds on how well graph topology alone enables link prediction for a wide variety of real-world graphs. We find that in the sparsest of these graphs the upper bounds are surprisingly low, thereby demonstrating that prediction systems on sparse graph data are inherently limited and require information in addition to the graph topology.

## 1 Introduction

Graph-based link prediction systems are widely used to recommend a wide variety of products and services. Whenever a shopping website predicts what you will buy next based on what you and others like you have previously bought, that's link prediction. Whenever a social media network suggests that you might know someone, that's link prediction. Link prediction is the well-studied task of predicting connections between entities amidst a network (aka "graph") of entity-entity connections.

Usually, these recommendations are based on a combination of node features and the topology (*i.e.* the link-structure) of the graph-data. One might assume that the node features or the topology contain sufficient information for an ideal link prediction system using that information to perfectly select the missing connections. However, we show this to be false for graph topology; whenever a graph possesses symmetries (*i.e.* automorphisms) in its topology, then the graph's topology does not contain enough information to guarantee correct selection of the missing edges. This raises two foundational questions in machine learning on graphs:

1. What are the inherent limits on a graph structure's predictability?

2. Do contemporary systems approach these performance limits?

The goal of the present work is to help answer these questions. To do so, we investigate how much information a graph's topology alone can provide to a link prediction algorithm; that forms one kind of inherent limit on a structure's predictibility. In particular, we calculate hard upper limits on how well an algorithm using only topology information can score in standard link prediction metrics (AUC and AUPR). Our upper bounds hold for *any and all* link prediction algorithms and thus are bounds on the informativeness of the topology data itself. Given these limits, we and others can begin to answer question 2 by comparing contemporary link prediction systems' scores to these upper bounds.

To calculate an upper bound that will hold regardless of the link prediction algorithm used, we calculate an upper limit on the performance of an idealized algorithm presented with as much information about the

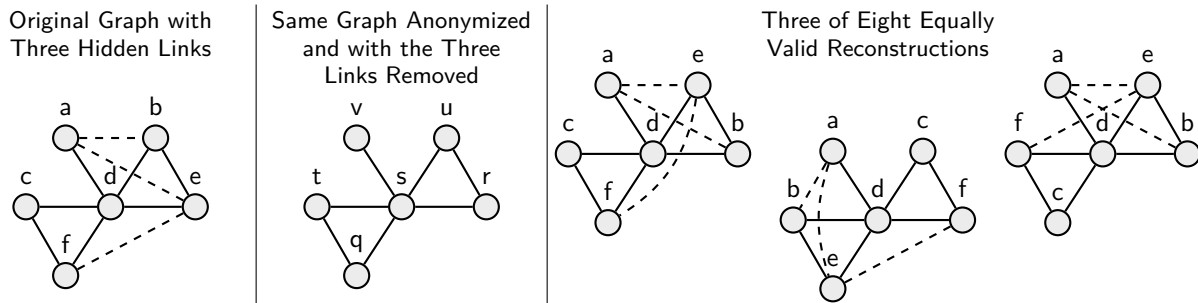

Figure 1: Toy example graph with three held-out edges for what we call the known-topology link prediction task. The link predictor knows that it must pick edges $(a, b)$, $(a, e)$, and $(e, f)$, but does not know for certain which nodes in the anonymized graph correspond to $b$, $c$, $e$, or $f$. In this case, there exist eight equally plausible edge sets that would turn the anonymized graph into a graph isomorphic to the original. However, only one of these eight is "correct" from the perspective of standard link prediction evaluation. Note that each of these eight candidate edge sets correspond to a different interpretation of the node labels in the anonymized graph.

solution as possible. Specifically, we imagine showing the algorithm the graph *and* the solution set of missing edges; then we imagine removing or scrambling the nodes' labels and asking the algorithm to predict the solution on the re-labeled graph. We call the process of removing or randomly permuting the nodes' labels "anonymizing" the graph.

For example, consider the following scenario illustrated in Fig. 1: Imagine you are shown both a link prediction task and the answer to the same task (*i.e.* the links one must predict); in Fig. 1, the solution edges are $(a, b)$, $(a, e)$, and $(e, f)$. Now imagine further that you are asked to perform link prediction on an anonymized version of the same problem, then asked to predict which edges are missing. You know, for instance, that you must select edge $(e, f)$, but you do not know for certain which node is $e$ nor which node is $f$.

The anonymized graph is, by itself, effectively identical to the data that an algorithm would be given for the regular link prediction task. We call the task of doing link prediction on the anonymized graph after having seen the un-anonymized solution *the known-topology link prediction task*. As we hinted above, and as Figure 1 shows, *symmetries* in the graph can render several possible edge-predictions structurally identical and therefore equally valid. It is from these symmetries that performance is limited even on the known-topology task.

Of course, in practice, most systems will perform much worse at the standard link prediction task than an ideal algorithm could perform at the known-topology task. Any link prediction algorithm will have some inherent, implicit modeling assumptions about the graph. For example, a simple model like triadic closure assumes that the likelihood of an edge is proportionate to the number of triangles the edge would be involved in (Bianconi et al. (2014); Klimek & Thurner (2013); Jin et al. (2001); Davidsen et al. (2002)). Expressed in a Bayesian fashion, we can think of a link prediction algorithm as conditionalizing on evidence, where the data the algorithm sees is its evidence and the algorithm's inherent assumptions form its prior. One could think about an algorithm performing the known-topology link prediction task as an algorithm doing standard link prediction with a perfect prior (*i.e.* 100% confidence on the correct graph). We focus on the known-topology link prediction task because it enables us to establish limits that exists in the data itself, without making *any* assumptions about the link prediction algorithm.

After discussing some related work in Section 2 and defining our formalisms in Section 3, we discuss some formal properties of link predictors as well as quantitative link prediction evaluation metrics (ROC and AUPR) in Section 4. Section 5 provides a formula for taking any specific link prediction task (*i.e.* any (graph, missing edge set) pair) and calculating the max possible scores a link predictor could achieve on the known-topology version of the task, thereby establishing an upper bound on performance for the standard

version of the task. We leave the proof that our formula gives the maximum score in the Appendix. In Section 6 we introduce our experiments, which consist of calculating link prediction score limits for various real-world graphs; this includes both our main known-topology task limit *and* various lower limits that appear on the known-topology task when the link prediction algorithm only uses a subset of its available information (a *k*-hop neighborhood) to predict whether a link will appear. Section 7 shows our results. We find that on sparse graphs commonly used for link prediction, the limits are surprisingly low. Then in Section 7.2 we observe how a famous graph neural network's reported results are *above* the performance limits and discuss how this is likely due to a common mistake in correctly measuring AUPR. Lastly, we discuss ways to generalize our analysis in Section 8.

## 2 Related Work

### 2.1 Link Prediction

The link prediction task is widely studied, almost to the point of being ubiquitous for graph research. It was introduced by Liben-Nowell & Kleinberg (2003). Given a graph, the task is to predict which edges (*i.e.* links) might be missing, either because the data is incomplete or because more edges will appear in the future.

Early models for link prediction tended to rely on certain assumptions, such as that of "triadic closure," the assumption that if A connects to B and B connects to C, then A is also likely to connect to C (Jin et al. (2001); Davidsen et al. (2002); Klimek & Thurner (2013); Bianconi et al. (2014); Adamic & Adar (2003)).

Then, following the historical trend of machine learning in general, link prediction models grew in complexity both in their features and in their classifiers (Al Hasan et al. (2006); Martínez et al. (2016)) until neural networks eventually began to outperform other methods (Zhou et al. (2020); Wu et al. (2020)).

### 2.2 Predictability Limits

To our knowledge, this is the first work to quantify a maximal performance score that link predictors can obtain on a given task. Other work in predictibility has focused on measures of predictibility distinct from evaluation scores. For instance, Abeliuk et. al. study how the predictability of time-series data degrades as the amount of data available decreases; they quantify predictability in terms of permutation entropy and signal self-correlation, as well as actual prediction performance of specific models (Abeliuk et al. (2020)). Permutation entropy has also been found to be useful to measure predictability in ecology and physics, and self(auto)-correlation in finance (Abeliuk et al. (2020); Bandt & Pompe (2002); Garland et al. (2014); Lim et al. (2013)).

Scholars have also analyzed predictability limits in other domains. For instance, some have used notions of entropy to measure predictability limits on human travel (Lu et al. (2013); Song et al. (2010)) and disease outbreaks (Scarpino & Petri (2019)). Predictability is related to system complexity and chaos (Boffetta et al. (2002)). For instance, minute uncertainties on initial conditions can greatly limit one's ability to make accurate weather forecasts (Zhang et al. (2019)).

Others have done excellent work on the related but distinct case that the ground truth (*i.e.* correct output) itself is uncertain or inherently fuzzy. For instance, in these sorts of settings one might need an alternate way of scoring a classifier, such as Survey Equivalence (Resnick et al. (2021)). Rather than fuzzy ground-truth, the present work focuses on cases where the correct output is clearly known during evaluation, but where limits in predictibility come from symmetries within the input data.

# 3 Formalisms

## 3.1 Graphs

We represent a graph $G$ as $G = (V, E)$ where $V$ is the set of vertices (*i.e.* nodes) and $E$ is the set of edges. The edges are pairs of vertices. If the graph's connections are considered to have a direction, we say that $E \subseteq V \times V$ and that the graph's *non-edges* are $(V \times V) \setminus E$. If the connections do not have a direction, then the edges are unordered pairs: $E \subseteq \{\{a, b\} \mid (a, b) \in V \times V\}$. However, for simplicity, it is standard to always write $(a, b)$ rather than $\{a, b\}$ even when talking about undirected graphs. An edge of the form $(a, a)$ is called a *self-loop*.

## 3.2 Isomorphisms

Given two graphs $G_1 = (V_1, E_1)$ and $G_2 = (V_2, E_2)$, we say that they are *isomorphic* if there exists a way to align the two graphs' vertices so that the structures overlap perfectly. Formally, $G_1$ and $G_2$ are isomorphic (expressed as $G_1 \cong G_2$) if there exists a bijection between the vertices $f : V_1 \to V_2$ such that $(a, b) \in E_1 \leftrightarrow (f(a), f(b)) \in E_2$. In this case the function $f$ is called an *isomorphism*. In this paper, whenever we refer to two graphs as being *equivalent* or *identical* we mean that they are isomorphic.

If $f$ is an isomorphism between two graphs $G_1$ and $G_2$ we will sometimes denote this as $G_1 \cong_f G_2$.

## 3.3 Automorphism Orbits

Within the context of a single graph, the *automorphism orbit* of an object (*i.e.* a vertex or an edge) captures its equivalence with other objects in the graph. Two objects are in the same orbit if and only if the data *in no way* distinguishes between the two objects.

An *automorphism* of a graph is an isomorphism of the graph with itself. That is, an automorphism of a graph $G = (V, E)$ is a bijective function $f : V \to V$ such that:

$$(a, b) \in E \leftrightarrow (f(a), f(b)) \in E$$

The set of all automorphisms of a graph $G$ form the *automorphism group* of the graph and is denoted $\text{Aut}(G)$.

The *automorphism orbits* of a graph typically refer to collections of equivalent vertices; however, they can also refer to collections of equivalent edges. The orbit of a vertex $a$ in graph $G$ is the set $\text{AO}_G(a) = \{f(a) \mid f \in \text{Aut}(G)\}$. Similarly, the orbit of an edge $e = (a, b)$ in graph $G$ is the set $\text{AO}_G(e) = \{(f(a), f(b)) \mid f \in \text{Aut}(G)\}$. Note that $a \in \text{AO}_G(a)$ and $e \in \text{AO}_G(e)$ due to the trivial automorphism $f(x) = x$.

We can even consider the orbits of *non-existent* edges (*i.e.* non-edges). Let $(a, b) \notin E$ be an edge which is not in $G$. We can still define the orbit of $(a, b)$ to be $\{(f(a), f(b)) \mid f \in \text{Aut}(G)\}$. These orbits are collections of edges not in $G$ which are equivalent to each other given $G$.

## 3.4 Induced Subgraphs

Given graph $G = (V, E)$ and a subset of the graph's vertices $S \subseteq V$, we can define $G$'s *induced subgraph* on $S$ to be a graph with $S$ as its nodeset and the edges in $G$ that connect nodes in $S$. Formally: $G(S) = (S, \{(a, b) \mid (a, b) \in E \wedge a \in S \wedge b \in S\})$.

## 3.5 K-hop Walks

Given two vertices $x, y \in V$, a *k-hop walk* from $x$ to $y$ is a sequence $\langle w_0, w_1, w_2, ..., w_{k-1}, w_k \rangle$ where $x = w_0$, $y = w_k$, and $(x_{i-1}, x_i) \in E$ for all $i$ from 1 to $k$. For convenience, we define a "zero-hop" walk to be a single-node "sequence" $\langle w_0 \rangle$, representing a "no-steps-taken" journey from $x$ to itself.

### 3.6 K-hop Neighborhoods

In practice, most link prediction algorithms do not use the entire graph when predicting the probability of edge membership. Rather, they tend to use local context. We formalize one intuitive notion of local context here that will be used throughout the paper.

Given a node or an edge, we can consider the nodes surrounding the entity to be the collection of nodes you could reach by beginning at the node (or the edge's endpoints), and taking up to $k$ steps (aka "hops") across edges for some value $k$. We can express this formally as follows:

Given a node $x \in V$, we define its *k-hop neighborhood nodes* $N_k(x)$ to be the set of all nodes within $k$ or fewer steps of $x$. Formally, $N_k(x) = \{y \mid \text{There exists an } l\text{-hop walk from } x \text{ to } y \text{ where } l \leq k\}$.

When we consider an edge or a non-edge $(a, b)$, we define its $k$-hop neighborhood nodes to be the union of the two endpoints' $k$-hop neighborhood node sets: $N_k((a, b)) = N_k(a) \cup N_k(b)$.

Finally we can define the *k-hop neighborhood subgraph* (or simply "$k$-hop neighborhood") for an edge or non-edge $e = (a, b)$. It is the induced subgraph on $e$'s $k$-hop neighborhood nodes: $G_k(e) = G(N_k(e))$.

### 3.7 Anonymized Graphs

Given a graph $G = (V, E)$, an *anonymized version* of $G$ is another graph $H$ isomorphic to $G$ with no particular relation between $G$'s node labeling and $H$'s node labeling. More specifically, to get an anonymized copy of $G$, you can select a random permuation $\pi : V \to V$; your anonymized graph is then $H = (V, \{(\pi(a), \pi(b)) \mid (a, b) \in E\})$.

### 3.8 Canonical Forms

A canonical form of a graph $G$ is a representation of $G$ that is produced in a way invariant to the node ordering of $G$. The idea of a canonical form is used by practical graph isomorphism algorithms such as Nauty and Traces (McKay & Piperno (2014)); they work by first converting two graphs $G_1$ and $G_2$ to canonical forms $C_1$ and $C_2$ respectively, then perform a trivial check to see if $C_1$ and $C_2$ are identical.

In other words, canonical forms are defined in terms of the algorithm that creates them. An algorithm $A$ produces canonical forms for graphs if, for all pairs of graphs $G$ and $H$, $A(G) = A(H)$ if and only if $G$ is isomorphic to $H$. The canonical form of $G$ with respect to $A$ is $A(G)$.

## 4 Link Predictors and Their Evaluation

### 4.1 Link Predictors

A link predictor is essentially a binary classifier for non-edges. It produces a verdict indicating whether the (non-)edge is or should be a member of the graph or not.

Let $G = (V, E)$ be a graph and $\bar{E}$ be the set of non-edges in $G$; that is, $\bar{E} = \{(a, b) \mid a, b \in V \wedge (a, b) \notin E\}$. A hard link predictor (*i.e.* hard binary classifier) for $G$ and $\bar{E}$ is a function $\ell : \bar{E} \to \{\texttt{Positive}, \texttt{Negative}\}$ that gives a non-edge a label (Positive/Negative). A soft link predictor (*i.e.* soft binary classifier) for $G$ and $\bar{E}$ is a function $\ell : \bar{E} \to \mathbb{R}$ that gives a non-edge a score. The higher the score, the more likely the non-edge is considered to be one of the Positives; the lower the score, the more likely the non-edge is considered to be a Negative. The function may be the result of training a model on a collection of correct edges/non-edges via manual parameter tuning, statistical analysis, or any number of other methods.

In practice, soft classifier scores are often turned into hard labels by picking a threshold value $t$ and giving all entities with a score $\geq t$ the Positive label and all others the Negative label.

### 4.1.1 Our Assumptions

For convenience in our subsequent analysis, we make an assumption about how link predictors operate. However, we also explain why a performance bound for this kind of link predictor is ultimately a bound on all link predictors.

Our key assumption is that whenever a link predictor uses graph topology information $I$ to give an edge $e$ a score, it would have given edge $e$ the exact same score if any of the graph nodes in $I$ had been labeled differently. In other words, the predictor's output is permutation-invariant on its input.

At first glance, this might sound like a big assumption, but there are only two ways that an algorithm which is not permutation-invariant can get a better score than an algorithm that is permutation-invariant:

- Case 1: The input graph's node ordering was based on some property of the solution graph.

- Case 2: The input ordering had no significance, but the algorithm gets lucky arbitrarily due to the input (and would have performed worse given a different arbitrary node input ordering).

Concerning Case 1: Nobody should want algorithms that make use of the kind of data in Case 1, because that data is not available in real-world link prediction settings (*e.g.* a sales website cannot use a node ordering based on what products you *will* buy).

Concerning Case 2: The possibility that an algorithm could get lucky is not a meaningful counter-example to a performance limit[1].

Consequently, our performance bound is effectively a bound for all link prediction algorithms - not just permutation-invariant ones.

For those who find these concepts interesting, we note that the notion of permutation invariance has been explored in the graph neural network (GNN) literature. For example, see the seminal work of Haggai Maron, who studies what permutation-invariant architectures enable networks to do (Maron et al. (2019b;a; 2018)). Much of the GNN research is focused on limits that different architectures impose or what architectures enable – for example, that simple message passing has a power equivalent to the 1-dimensional Weisfeiler Lehman algorithm (Morris et al. (2019)) – whereas our work in this paper asks what limits are in the data itself regardless of which architecture is used.

### 4.1.2 Edge Equivalence

Our assumption from Section 4.1.1 could be rephrased as, "If a link predictor is given the exact same information about two different non-edges (same up to isomorphism), then it gives those two edges the same score." Using this assumption, we can partition edges into cells based on whether or not a link predictor is given the same information about the edges – same information $\leftrightarrow$ same cell.

When a link predictor $\ell$ for a graph $G$ uses the context of the entire graph to give an edge a score, then two edges are guaranteed to get the same score if $G$'s topology in no way distinguishes between the two. Formally, this means that two edges are guaranteed to get the same score if they are in the same automorphism orbit: $e_1 \in \mathrm{AO}_G(e_2) \rightarrow \ell(e_1) = \ell(e_2)$

Often, link predictors use a local context surrounding an edge to give it a score rather than using the entire graph as context. If we assume that a link predictor uses at most the $k$-hop neighborhood surrounding an edge to give the edge its score, then we get that when two edges have equivalent $k$-hop neighborhoods and

---

[1]Technically, there is one counter-example to this claim which might be of interest to the theoretically-minded reader. Due to non-linearities in how link prediction scores are calculated, an algorithm that coordinates its predictions across edges might manage to increase the expected value of its link prediction score even though in some cases it will still perform worse than a permutation-invariant algorithm. For example, in Figure 1 an optimal algorithm meeting our assumptions would give edges $(v, u)$, $(v, r)$, $(v, t)$, and $(v, q)$ in the anonymized graph each a $\frac{1}{2}$ probability score of being an edge. A non-permutation-invariant algorithm could give edges $(v, t)$ and $(v, q)$ each a probability score of 1 and edges $(v, u)$, $(v, r)$ a score of zero; this kind of prediction would do better half the time (*i.e.* on half the anonymizations) and worse half the time, but might have a higher expected performance score overall. We consider this sort of case to be exotic enough that it is not relevant to our analysis.

when those edges have the same role within the $k$-hop neighborhoods, then the edges get the same score. Formally, this can be expressed as: $(\exists f.\ G_k(e_1) \cong_f G_k(e_2) \wedge f(e_1) = e_2) \rightarrow \ell(e_1) = \ell(e_2)$

Note that by definition $e_1 \in \mathrm{AO}_G(e_2)$ implies $(\exists f.\ G_k(e_1) \cong_f G_k(e_2) \wedge f(e_1) = e_2)$ for any $k$. In other words, if two edges are equivalent in the context of the entire graph, then they are guaranteed to be equivalent when considered in context of their $k$-hop neighborhoods.

### 4.2 Performance Scores for Link Predictors

In practice, almost all link prediction classifiers are soft classifiers. There are a number of nuances to how these classifiers are scored that are worth highlighting here.

Remember that a soft classifier can be converted into a hard (i.e. binary) classifier by predicting "yes" when the soft classifier's output is above a certain threshold and "no" otherwise. Researchers tend to evaluate the soft predictors across a range of different thresholds. Each (soft predictor, threshold) pair represents a possible hard link predictor. Thus performance of a soft predictor can be considered to be the goodness of the collection of hard predictors it offers. This can be measured in terms of different criterion. One common criterion is the relationship between a predictor's True Positive Rate (TPR) and False Positive Rate (FPR), which generates the widely used ROC curve; the ROC score is the area under the ROC curve. Another common criterion is the relationship between the predictor's Precision and Recall, which leads to the Precision-Recall curve and its corresponding metric of Area Under the Precision-Recall curve (AUPR). For an in-depth analysis exploring the relationship between ROC curves and Precision-Recall curves, we recommend the paper by Davis & Goadrich (2006).

When converting a set of (TPR, FPR) or (Precision, Recall) points into a curve, an interpolation between points represents a way of combining the two hard classifiers (the two points) into a new hard classifier. This can be done by picking a value $\alpha \in [0, 1]$ and tossing an $\alpha$-weighted coin every time an entity is scored to decide which of the two hard classifiers to use for the entity. This is implicitly how we as well as Davis and Goadrich perform interpolation. It turns out that the popular trapezoidal interpolation is incorrect for Precision-Recall space, because hard classifiers cannot be combined to get precision-recall pairs that interpolate linearly (Davis & Goadrich (2006)).

Sometimes, rather than calculate the AUPR curve exactly, it can be approximated with a measure called Average Precision (AP). Rather than doing a complex interpolation between two precision-recall points, Average Precision simply uses the precision of the rightmost point (the point with the higher recall).

## 5 Optimal Prediction Performance

Here we offer the actual formulae for calculating the optimal ROC and AUPR scores a soft classifier can obtain.

Recall from Section 4.1.1 that there will be some edges which are topologically identical to each other and thus which will each get the same score from a link predictor. This fact forms a limit on the ability of the link predictor to separate positive edges from negative edges. As we discussed in Section 4.1.2, given whatever topological information an algorithm uses, we can partition the edges into cells, where each cell is one of the sets of edges that must all be given the same score as each other by the algorithm; when the algorithm uses the entire graph as data to help it score an edge, then the relevant partition is formed by grouping the edges according to their automorphism orbits.

Given a graph $G = (V, E)$ and its non-edges $\bar{E}$, as well as some link-prediction algorithm, we can consider the relevant partitioning of $\bar{E}$ into $k$ cells $C_1, C_2, ..., C_k$. A cell might contain only positive edges (*i.e.* only edges the link predictor should give a high score to), only negative edges, or a mixture. It's from mixed cells that performance limits arise: since all elements in a mixed cell get the same score, they cannot all have correct scores. For a cell $C_i$, we denote the number of positives in the cell as $p_i$, the number of negatives in the cell as $n_i$, and the the total number of elements in the cell as $t_i = p_i + n_i = |C_i|$.

For a given partitioning $C_1$ through $C_k$, we prove in the Appendix that the optimal ROC and AUPR scores a soft classifier can obtain equals the ROC/AUPR scores obtained from a classifier $\ell : \bar{E} \to \mathbb{R}$ which satisfies the following property:

$$\forall 1 \leq i, j \leq k. \ \forall e \in C_i, \ e' \in C_j. \ \left( \ell(e) \geq \ell(e') \right) \leftrightarrow \left( \frac{p_i}{t_i} \geq \frac{p_j}{t_j} \right) \tag{1}$$

Note that within a cell $C_i$, the probability that an element is a positive is just $\frac{p_i}{t_i}$. Thus a classifier that scores non-edges with the probability they're positives will get an optimal score – optimal given the topological information that the algorithm uses to distinguish non-edges.

This property of optimal classifiers permits us to easily compute the maximal ROC/AUPR scores that any algorithm could have obtained on a given dataset and task. All we need to do is find the partitioning of non-edges into cells that are topoligically equivalent given the data the link predictor makes use of. Then we order those cells according to their density of positives (*i.e.* according to $\frac{p_i}{t_i}$). Given that ordering, we use the standard ROC and AUPR formulae to calculate what scores would be obtained by an optimal classifier. We provide code both for proper AUPR calculation and for optimal ROC/AUPR scores at `https://github.com/SteveWillowby/Link_Prediction_Limits`.

For the formulae below, assume that the cells $C_1$ through $C_k$ are ordered such that $i \leq j \to \frac{p_i}{t_i} \geq \frac{p_j}{t_j}$.

To give the ROC and AUPR formulae in terms of this partitioning, we need just a bit more notation. Define cumulative sums $P_0 = 0$ and $P_i = \sum_{j=1}^{i} p_j$ for $1 \leq i \leq k$. Similarly, define cumulative sums $T_0 = 0$ and $T_i = \sum_{j=1}^{i} t_j$ for $1 \leq i \leq k$. And again, $N_0 = 0$ and $N_i = \sum_{j=1}^{i} n_j$ for $1 \leq i \leq k$. Define $T = T_k$, $P = P_k$, and $N = N_k$. Note that $|\bar{E}| = T = N + P$ (total number of non-edges = total number of things classified = negatives + positives). We now get the following formula for ROC:

$$\text{Max ROC} = \sum_{i=1}^{k} \frac{p_i}{P} \cdot \frac{N_i + N_{i-1}}{2N} \tag{2}$$

The formula for AUPR is messier due to the need for proper interpolation between precision-recall points discussed above in Section 4.2, but it is still easy to calculate:

$$\text{Max AUPR} = \sum_{i=1}^{k} \frac{p_i}{P} \cdot \frac{p_i}{t_i} \cdot \left( 1 + \left( \frac{P_{i-1}}{p_i} - \frac{T_{i-1}}{t_i} \right) \cdot \ln \left( \frac{T_i}{T_{i-1}} \right) \right) \tag{3}$$

Note that there are no division-by-zero issues with this formula due to the following facts: When $p_i = 0$, the entire expression becomes zero. Further $t_i$ always $> 0$. Lastly, when $T_{i-1} = 0$, then $P_{i-1} = 0$, and because $\lim_{x \to 0^+} x \ln \frac{1}{x} = \lim_{x \to 0^+} x \left( \ln(1) - \ln(x) \right) = 0$, we do not get a division by zero issue with $T_{i-1}$.

Equipped with these formulae, we can now begin to calculate the maximum possible performance scores on actual prediction tasks.

As we mentioned above, Average Precision (AP) is sometimes used to approximate AUPR. However, the nice result we prove for ROC and AUPR concerning the edge partioning order does *not* hold for AP. Fortunately, our upper bound on AUPR is also an upper bound on AP, so we can still upper-bound the AP scores that one might obtain. We provide a short proof of this in the Appendix.

## 6    Methodology

Our main experiment is to calculate how maximum link prediction scores vary with the amount of information given to an idealized algorithm. We run this test on a wide variety of real-world graphs. The procedure runs as follows:

| Graph | Dir | Weighted | $|V|$ | $|E|$ | # SL | AD | CC | Diam | ASP |
|---|---|---|---|---|---|---|---|---|---|
| Species 1 Brain [1] | D | U | 65 | 1139 | 0 | 35.0 | .575 | 4 | 1.83 |
| Highschool Friendships [2] | D | W | 70 | 366 | 0 | 10.5 | .362 | 12 | 3.90 |
| Foodweb [2] | D | U | 183 | 2476 | 18 | 27.1 | .173 | 6 | 1.84 |
| Jazz Collaboration [2] | U | U | 198 | 5484 | 0 | 55.4 | .617 | 6 | 2.22 |
| Faculty Hiring (C.S.) | D | W | 206 | 2929 | 124 | 28.4 | .214 | 7 | 2.88 |
| Congress Mentions [2] | D | W | 219 | 586 | 2 | 5.35 | .160 | 12 | 4.48 |
| Medical Innovation [2] | D | U | 241 | 1098 | 0 | 9.11 | .210 | 9 | 3.26 |
| C-Elegans Metabolic [2] | U | U | 453 | 2025 | 0 | 8.94 | .646 | 7 | 2.66 |
| USA Top 500 Airports (2002) [3] | D | U | 500 | 5960 | 0 | 23.8 | .617 | 7 | 2.99 |
| Eucore Emails [4] | D | U | 1005 | 24929 | 642 | 49.6 | .366 | 7 | 2.65 |
| Roget Concepts [3] | D | U | 1010 | 5074 | 1 | 10.0 | .108 | 14 | 4.89 |
| CCSB-YI1 [5] | U | U | 1278 | 1641 | 168 | 2.57 | .045 | 14 | 5.36 |
| MySQL Fn. Calls [6] | D | U | 1501 | 4212 | 13 | 5.61 | .078 | 18 | 5.36 |
| USA Airports (2010) [3] | D | U | 1574 | 28236 | 0 | 35.9 | .489 | 9 | 3.20 |
| Collins Yeast [3] | U | U | 1622 | 9070 | 0 | 11.2 | .555 | 15 | 5.53 |
| Cora Citation [1] | D | U | 2708 | 5429 | 0 | 4.01 | .131 | 15 | 4.53 |
| Citeseer Citation [1] | D | U | 3264 | 4536 | 0 | 2.78 | .072 | 10 | 2.64 |
| Roman Roads (1999) [3] | D | U | 3353 | 8870 | 0 | 5.29 | .025 | 57 | 25.3 |
| USA Powergrid [3] | U | U | 4941 | 6594 | 0 | 2.67 | .080 | 46 | 19.0 |

Table 1: Graphs used for tests – The edge count does not include self-loops, which are listed separately – Key: Dir = (**Un**)**D**irected, Weighted = (**Un**)**W**eighted, # SL = # Self-loops, AD = Average Degree, CC = Average Clustering Coefficient, Diam = Diameter, ASP = Average of all Shortest Path Lengths –
Sources: [1]: Rossi & Ahmed (2015), [2]: Kunegis (2013), [3]: Clauset et al. (2016), [4]: Leskovec & Krevl (2014), [5]: Yu et al. (2008), [6]: Myers (2003)

1. Begin with a graph $G = (V, E)$ and an edge removal probability $p$ (we set $p \leftarrow 0.1$).

2. Define the set of negatives $N$ as all edges not in $G$.

3. Remove each edge in $G$ with probability $p$ (independently) and add the removed edges to the set of positives $P$. Call the resulting graph $H \leftarrow (V, E \setminus P)$.

4. Get a (hashed) canonical representation for each non-edge's automorphism orbit in $H$.

5. Use the collected information to calculate the maximum scores via equations 2 and 3.

6. Assign $k \leftarrow 1$.

7. Get a (hashed) canonical representation of the $k$-hop neighborhood for each non-edge in $H$ where the non-edge's endpoints are given a distinct color from the rest of the nodes.

8. Use the collected information to calculate the maximum scores when using at most $k$ hops of information about a non-edge.

9. If the performance limit just obtained from step 8 is equal to (or within 0.005 of) the performance limit obtained from step 5, then stop. Otherwise, assign $k \leftarrow k + 1$ and go to step 7.

We perform the above procedure multiple times for each graph. Each iteration corresponds to different, random possible sets of missing edges; each set of missing edges can be slightly different in terms of the limit on its predictability. We get the mean value and 95% confidence interval for each distinct value of $k$.

We tested the link prediction limits on a wide variety of real-world graphs. They are listed in Table 1.

# 7 Results

## 7.1 Sparsity Tends to Lower the Upper-Bound

We found that on most graphs, the upper bounds were near 100%, even when using 1-hop neighborhoods; we suspect that this is because when degrees are high enough there is still a large number of possible 1-hop neighborhoods such that the hypothetical optimal algorithm can take advantage of the slightest difference between neighborhoods. However, we found that on the sparsest graphs the results told a different and very interesting story.

We show the results for the four sparsest graphs: the Cora and Citeseer citation (sub)graphs, the CCSB-YI1 Protein-Protein Interaction graph, and a US Powergrid network. The results are in Figure 2. In particular, we focus on the AUPR values, because even though link prediction papers often report ROC scores, link predictors can easily get large ROC scores due to the class imbalance (the sheer number of non-edges) (Yang et al. (2015)).

In summary, our results give good evidence that when data becomes sparse enough, graph toplogy alone is severely limited in its ability to indicate a difference between genuine and fake missing edges.

## 7.2 Negative Sampling Methodologies Produce Artificially High Scores

We were curious to see how these fundamental limits compared to reports of link prediction performance. As a small case study, we considered the seminal Graph Convolutional Neural Network Auto-Encoder (GCNAE), a widely used and referenced model that can perform topology-only link prediction (Kipf & Welling (2016)). This will help us discern how well link predictors are making use of the topology information available to them. Though this model is a little bit "old" in the fast-paced world of graph neural networks, its performance is still within just a few percentage points of modern GNN performance (Chen et al. (2020); Fey et al. (2021)). In the original GCNAE paper, the authors tested their model on undirected versions of the Cora and Citeseer citation networks. They reported ROC scores and AP scores.

We found that the AP scores they reported for the Citeseer network were well *above* our upper bound, indicating that there was a difference in the calculated AP. To understand this, we looked at the GCNAE code and found that in its tests the number of negative edges was downsampled to one negative test edge per positive test edge. This sort of downsampling is common when performing link prediction evaluation with AP or AUPR; however downsampling tends to boost the AP and AUPR scores significantly relative to what they would have been if the full set of negatives was used in testing (Yang et al. (2015)). The GCNAE paper itself does not specify that downsampling occurred.

By contrast, the paper's reported ROC scores are well below our upper bound on ROC. This makes sense as the ROC score is not affected by downsampling (Yang et al. (2015)). If we downsample the number of negative edges to one negative edge per positive edge when calculating the AP limits, we get that the GCNAE's AP performance is also well below the upper bound. We show the numeric results in Table 2.

The point of this case study is twofold. Firstly, a metric (*e.g.* AP) may have different meanings depending on how it is used, and our methodology may be able to help retroactively determine which approach was used if the original paper does not specify.

Secondly, and perhaps of greater interest, state of the art link prediction systems using the topology of a network do not reach the topology-based upper limit on performance. We take this to suggest either that state of the art link prediction systems have room for improvement in their use of graph topology *or* that what structurally differentiates the $k$-hop neighborhoods of true edges from the $k$-hop neighborhoods of false edges in our tests is basically noise that an algorithm should not pay attention to if it wishes to generalize well. After all, if for example the 1-hop neighborhoods of two different non-edges both have 20 edges per neighborhood and differ in only one place, should we expect a link prediction algorithm to always treat that difference as significant? We propose some future work in Section 8.1 for exploring how the upper limit on performance changes when the resolution of the data is a bit blurrier, thereby reducing this noise.

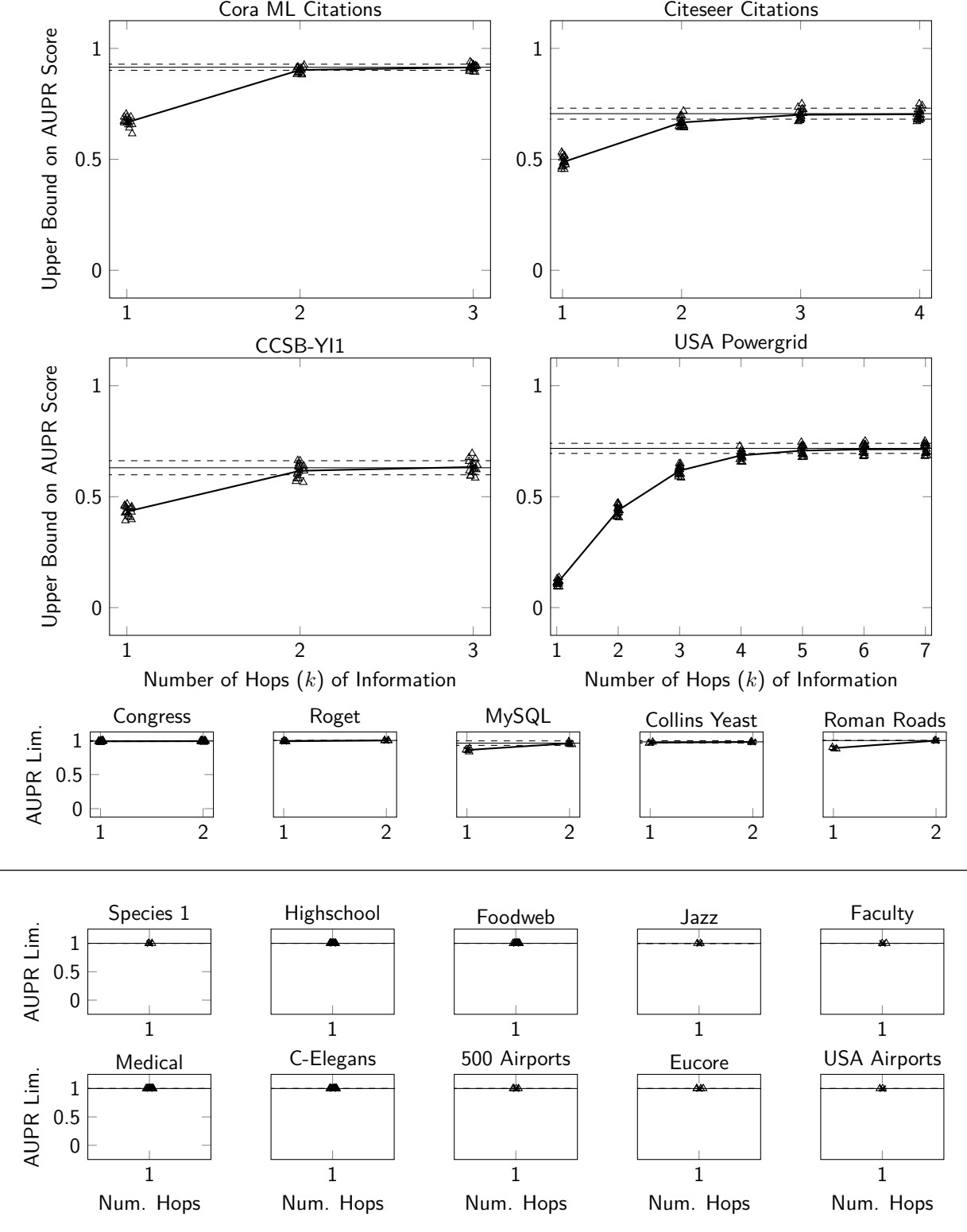

Figure 2: Hard upper bounds on link prediction performance as it varies with the amount of information given to a link prediction algorithm. The horizontal line shows the limit when using the entire graph ($k = \infty$). Ten percent of the graph's edges were randomly selected as test edges. The multiple points at a single value of $k$ are from different sets of randomly chosen test edges; left-right jitter is employed to aid in visualization. Error bars are 95% confidence intervals.

| Graph | GCNAE ROC | ROC Upper-Bound | GCNAE AP | AP Upper-Bound | AP Upper-Bound (1:1 Downsampling) |
|---|---|---|---|---|---|
| Cora | 0.843 ± 2e-4 | 0.99992 ± 3e-5 | 0.881 ± 1e-4 | 0.903 ± 0.020 | 0.99999 ± 9e-6 |
| Citeseer | 0.787 ± 2e-4 | 0.9981 ± 3e-4 | 0.841 ± 1e-4 | 0.686 ± 0.019 | 0.9989 ± 5e-4 |

Table 2: Comparison to the GCNAE's Reported Results – The ± symbol indicates the 95% confidence interval. We conclude that the GCNAE paper is likely downsampling negative test edges in the process of calculating AP. More importantly, once downsampling is factored in, there is a notable gap between the hypothetical ideal performance and state of the art topology-based performance. We discuss this more in Sec. 7.2. Note: These results are for undirected versions of the graphs, whereas the results in Fig. 2 are for the directed versions (GCNAE only does link prediction on undirected graphs). Also, note that the version of Citeseer that the GCNAE paper used has some extra nodes with no links, whereas the version we used for Fig. 2 does not.

### 7.2.1 Confirming our Assumption

To verify that the discrepency between score and upper bound is indeed due to downsampling negative edges, we ran the GCNAE code and obtained link prediction scores when using all possible negative edges. The results confirmed our hypothesis: When using the full set of negative test edges, the average ROC scores we obtained for Citeseer and Cora (∼77% and ∼84% respectively) were similar to the paper's reported results, but the average AP scores were *much* lower (just ∼1.2% and ∼1.4% respectively). These extremely low scores do not mean that GCNAE is performing poorly, for true AP is a much harsher and informative link prediction metric than ROC, and other GNN models get similar scores on similar datasets (Yang et al. (2015); Hibshman et al. (2021)).

## 8 Discussion

### 8.1 Applications, Extensions, and Limitations

In addition to the fact that our methodology gives insights about topology-based link prediction, we believe the kind of analysis we offer in this paper can be extended and expanded. We observed how maximum possible performance on a particular binary classification task (*i.e.* link prediction) varies with the amount of information available to the classifier. At a certain resolution, inputs to the algorithm look identical. In our analysis, the differing resolutions were the differing $k$ for the $k$-hop neighborhood subgraphs. However, these resolutions could hypothetically be any reasonable representation of the data.

If these kinds of equivalence partitions can be created at widely varying resolutions for a classification tasks, then researchers will begin to be able to say things like "our algorithm works as well on the full data as an optimal algorithm would work on data of resolution $X$."

The key ingredient for our analysis was that at a given resolution we were able to partition the objects being classified (the non-edges) into cells of equivalent objects; this let us calculate how well a hypothetical optimal algorithm would perform on those cells. We were able to get this kind of partitioning because our equality relation on two test objects was isomorphic equivalence of the objects' $k$-hop neighborhoods *and thus the relation was transitive.* Yet we expect that even in cases where a transitive equality relation is not immediately available, one could create such a relation by using a distance measure to cluster test inputs and then defining equality as being in the same cluster. The more fine-grained the clusters, the higher the resolution of data given to the hypothetical optimal algorithm.

The main limit we are aware of for this kind of analysis is that at high data resolution, noise can easily dominate the analysis. That is to say, at high resolution, random noise tends to render each entity to be classified unique, and thus the hypothetical, optimal algorithm with a perfect prior will be able to correctly distinguish any two entities and get a perfect score.

For example, consider link prediction on pure noise: That is, consider the process of randomly generating a graph where each edge is present independently with some probability $p$ and then randomly hiding some

fraction of the edges to create a link prediction task. Such a random graph will likely have no global symmetry (Erdős et al. (1960)), so at high data resolution (*e.g.* $k = 3$) every non-edge will be unique and thus the hypothetical, optimal algorithm with a perfect prior will obtain a perfect score, even though a real-world algorithm that does not mystically foreknow the answer can do no better than considering each edge to be equally likely, because that is in fact how the graph was constructed.

Fortunately for our kind of analysis, real-world algorithms are usually designed to ignore noise in the first place, so a data resolution that successfully filters out noise can simultaneously be relevant and provide a non-trivial upper bound on optimal performance.

## 8.2 Conclusion

We presented a methodology for calculating hard limits on how well a link prediction algorithm could perform when using structural information only. This helps analyze how much information graph structure does or does not provide for link prediction. We found that very sparse graphs give rise to significant inherent difficulties and therefore contain strong caps on optimal performance.

We also observed that a state of the art topology-based link prediction method performs well below the upper bound in some cases, which we believe either means that the link prediction algorithms have serious room for improvement *or* that our test sometimes picks up on "noise" that indeed differentiates edges from non-edges but which an algorithm should not be expected to pick up on because that noise would not behave in any consistent or infer-able manner. These observations prompted our discussion on further avenues of discovery and extensions of our methodology; we expect that an analysis similar to ours which finds a way to obtain performance upper bounds at varying degrees of blurring the noise would provide further insights.

**Acknowledgments**

This research is supported by grants from the US National Science Foundation (#1652492 and #1822939).

Thank you to the reviewers for their thoughtful feedback.

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

# A    Appendix - Proofs

## A.1    Maximum ROC

Let $k$ be the number of distinct inputs to the classifier (*i.e.* the number of cells of isomorphically-equivalent non-edge types in the partition of non-edges).

Further, let $s$ be a classifier that achieves an optimal ROC score for these cells. Because ROC is calculated with respect to the different (true positive rate, false positive rate) pairs obtainable by using different score thresholds to convert a soft classifier into various hard classifiers, then without loss of generality $s$ assigns a distinct score to each cell; if $s$ did not, then we could just consider the cells given the same scores as being the same cell with a larger total size and larger number of positives.

As in Section 5, let $t_1, t_2, ..., t_k$ be the total sizes of the cells, where the cells are ordered by the score $s$ gives to them ($t_1$ is the total size of the cell with the highest score). Likewise, let $p_1, p_2, ..., p_k$ be the number of positives in the respective cells. For notational convenience, we define $t_0 = p_0 = 0$.

Let $T_i = \sum_{j=0}^{i} t_j$, $P_i = \sum_{j=0}^{i}$, and $N_i = T_i - P_i$. Also define $T = T_k$, $P = P_k$, and $N = N_k$.

Assume for sake of contradiction that there exists an $i \in [k-1]$ such that $\frac{p_i}{t_i} < \frac{p_{i+1}}{t_{i+1}}$. Now imagine an alternate classifier $s^*$ that gives the exact same scores as $s$ except that it reverses the scores for the $i$'th and $(i+1)$'th cells.

Then we get a new set of variables $t_1^*, t_2^*, ..., t_k^*$ and $p_1^*, p_2^*, ..., p_k^*$ as well as corresponding $T_j^*, P_j^*$, and $N_j^*$ where the values correspond to the cells as ordered by classifier $s^*$. Due to the definition of $s^*$, $t_j = t_j^*$ and $p_j = p_j^*$ for all $j$ except $j \in \{i, i+1\}$, in which case $t_i = t_{i+1}^*$, $t_{i+1} = t_i^*$, $p_i = p_{i+1}^*$, and $p_{i+1} = p_i^*$. Once again, for notational simplicity we pad the beginning of these lists with $t_0^* = p_0^* = 0$.

This gives us a total of $k+1$ (true positive rate, false positive rate) pairs (i.e. (TPR, FPR) pairs). The false positive rate is the x axis of the curve and the true positive rate is the y axis.

$$\text{True Positive Rate } j \text{ for } s \text{ (i.e. TPR}_j) = \frac{P_j}{P} \text{ for } j \in \{0, 1, ..., k\}$$

$$\text{False Positive Rate } j \text{ for } s \text{ (i.e. FPR}_j) = \frac{N_j}{N} \text{ for } j \in \{0, 1, ..., k\}$$

Likewise for $s^*$ we have:

$$\text{True Positive Rate } j \text{ for } s^* \text{ (i.e. TPR}_j^*) = \frac{P_j^*}{P} \text{ for } j \in \{0, 1, ..., k\}$$

$$\text{False Positive Rate } j \text{ for } s^* \text{ (i.e. FPR}_j^*) = \frac{N_j^*}{N} \text{ for } j \in \{0, 1, ..., k\}$$

The ROC curve then interpolates linearly between these points. Note that by definition $\text{TPR}_0 = \text{TPR}_0^* = \text{FPR}_0 = \text{FPR}_0^* = 0$ and $\text{TPR}_k = \text{TPR}_k^* = \text{FPR}_k = \text{FPR}_k^* = 1$.

We can consider the interpolation between the $j$'th (TPR, FPR) point and the $(j+1)$'th (TPR, FPR) point as corresponding to a variable $\alpha \in [0, 1]$ where:

$$\text{TPR}_{j,\alpha} = \frac{P_j + \alpha p_{j+1}}{P}$$

$$\text{FPR}_{j,\alpha} = \frac{N_j + \alpha(t_{j+1} - p_{j+1})}{N}$$

This leads to the following:

$$\frac{d}{d\alpha} \text{TPR}_{j,\alpha} = \frac{p_{j+1}}{P}$$

and separately:

$$\alpha = \frac{N \cdot \text{FPR}_{j,\alpha} - N_j}{t_{j+1} - p_{j+1}}$$

$$\frac{d}{d\text{FPR}_{j,\alpha}} \alpha = \frac{N}{t_{j+1} - p_{j+1}}$$

Using the chain rule gives us:

$$\frac{d}{d\text{FPR}_{j,\alpha}} \text{TPR}_{j,\alpha} = \frac{d\alpha}{d\text{FPR}_{j,\alpha}} \cdot \frac{d\text{TPR}_{j,\alpha}}{d\alpha} = \frac{N}{P} \cdot \frac{p_{j+1}}{t_{j+1} - p_{j+1}}$$

For $s^*$'s curve this becomes:

$$\frac{\mathrm{d}}{\mathrm{dFPR}_{j,\alpha}^*}\mathrm{TPR}_{j,\alpha}^* = \frac{N}{P} \cdot \frac{p_{j+1}^*}{t_{j+1}^* - p_{j+1}^*}$$

Now, because $t_j$ and $t_j^*$ only differ at $j = i$ and $j = i+1$, and the same holds for $p_j$, etc., and because the values at $i$ and $i+1$ are the reverse of each other, then we can conclude that the $(i-1)$'th TPR-FPR point is the same for both $s$ and $s^*$ as well as the $(i+1)$'th point. The only difference is at the $i$'th point. To see that the $(i+1)$'th point is the same, note that $P_{i+1} = P_{i-1} + p_i + p_{i+1} = P_{i-1}^* + p_{i+1}^* + p_i^* = P_{i+1}^*$.

Further, because $\frac{p_i}{t_i} < \frac{p_i^*}{t_i^*}$ we obtain that $\frac{p_i}{t_i - p_i} < \frac{p_i^*}{t_i^* - p_i^*}$ which in turn means that:

$$\frac{\mathrm{d}}{\mathrm{dFPR}_{i-1,\alpha}}\mathrm{TPR}_{i-1,\alpha} < \frac{\mathrm{d}}{\mathrm{dFPR}_{i-1,\alpha}^*}\mathrm{TPR}_{i-1,\alpha}^*$$

In other words, the slope leading from the $(i-1)$'th point to the $i$'th point is greater in $s^*$'s ROC curve than in $s$'s. Furthermore, $\frac{\mathrm{d}}{\mathrm{dFPR}_{i-1,\alpha}}\mathrm{TPR}_{i-1,\alpha} = \frac{\mathrm{d}}{\mathrm{dFPR}_{i,\alpha}^*}\mathrm{TPR}_{i,\alpha}^*$, and $\frac{\mathrm{d}}{\mathrm{dFPR}_{i,\alpha}}\mathrm{TPR}_{i,\alpha} = \frac{\mathrm{d}}{\mathrm{dFPR}_{i-1,\alpha}^*}\mathrm{TPR}_{i-1,\alpha}^*$, meaning that $s^*$'s curve goes up more sharply first, and $s$'s curve catches up with the exact same slope later. Since both curves up to and including their $(i-1)$'th point are identical, and since they are also identical at the $(i+1)$'th point and thereafter, this means that $s^*$'s curve has a larger area underneath it.

Ergo, we obtain a contradiction, for $s^*$ obtains a higher ROC score than $s$. Thus the assumption must have been false. This means that $s$ orders the cells such that $\frac{p_j}{t_j} > \frac{p_{j+1}}{t_{j+1}}$ for all $j \in [k-1]$. In other words, $s$ completely sorts the cells as we intended to show. □

## A.2 Maximum AUPR

Davis and Goadrich have shown that if one ROC curve dominates another, then the corresponding (properly interpolated) precision-recall curves yield the same dominance (Davis & Goadrich (2006)). Thus the AUPR result follows directly from our ROC result above.

## A.3 Maximum AP vs. Maximum AUPR

The nice ordering property we prove for ROC and AUPR does not hold for AP. For example, consider the following three cells with number of (positives, negatives) total: $\langle (10,0), (2,2), (9,7) \rangle$. Ordering these in decreasing $\frac{\text{Positives}}{\text{Positives}+\text{Negatives}}$ yields an AP of approximately 0.856 whereas ordering them in the order listed yields an AP of approximately 0.858.

Fortunately, we can still calculate a hard upper limit on AP scores by simply using the upper limit on the AUPR score. Remember our ordering rule for an optimal curve: Order by $\frac{\text{Positives}}{\text{Positives}+\text{Negatives}}$ in a descending order. Remember also that via the ROC and AUPR proofs, we showed that given *any* AUPR curve obtained by some ordering of the cells where some pair of cells disobeyed our ordering rule, you could get an AUPR curve that *dominates* it by swapping the ordering of the incorrect pair. Now, observe that any curve which does not follow the ordering rule can be converted into the curve that does by a succession of swaps where the swaps are of adjacent, incorrectly ordered cells; this corresponds to the naive "bubble sort" algorithm (Astrachan (2003)). During this process of swaps, each new curve dominates the former. Given that the last curve is the one corresponding to our ordering, it follows that our optimal curve not only has a larger area underneath it than any other curve, but it also *dominates* any other curve. Last but not least, note that if you follow the optimal AUPR curve from right to left (*i.e.* from recall of 1 to recall of 0) that the curve never decreases in precision.

Now let us turn our attention to AP curves. Recall that an AP curve is obtained in a very similar manner to an AUPR curve. In both cases you first get a collection of precision-recall points given the ordering the classifier gives to the cells (we call these precision-recall points the "base points"); the only difference between AP and AUPR is in how the two kinds of curves interpolate between adjacent base points. AP

curves interpolate between any two precision-recall points by using the precision from the point with the higher recall.

We can observe two things: First, given our observations about the optimal AUPR curve, regardless of the ordering of the cells used to get the base points, all the base points of the AP curve are either on or are strictly dominated by the points in the optimal AUPR curve. Second, as you move from right to left along the interpolated AP points, the precision value remains constant until you hit a new base point. By contrast, as we discusse above the optimal AUPR curve's precision value is either constant *or increasing* as you move from right to left. Thus any AP curve is always dominated by the ideal AUPR curve. □

