# OpenReview forum: "Inherent Limits on Topology-Based Link Prediction"
_TMLR — Accepted by TMLR_

### Review · Reviewer_s32G · 2023-03-10

**Summary Of Contributions:**

This paper proposes hard upper bounds of link prediction performance based on topology, which brings new insight on analyzing the utilization of graph structure information for link prediction. The authors conduct experiments on a variety of real-world graphs to calculate how maximum link prediction scores vary with the amount of information based on proposed optimal prediction performance measurements. The experimental results demonstrate that link prediction methods on sparse graphs are inherently limited and require information besides the graph topology.

**Audience:**

Yes

**Broader Impact Concerns:**

None.

**Claims And Evidence:**

Yes

**Requested Changes:**

There are several suggestions:
1. The relations and potential impact on topology-beyond link prediction methods should be discussed in this paper.
2. More topology-based methods need to be considered to provide more convincing conclusions.
3. The article structure and writing need to be improved so that readers can understand the contribution of the article more easily.

**Strengths And Weaknesses:**

Strengths:

S.1 This paper provides a new approach to analyse topology-based link prediction methods and the upper bounds will help the evaluation of link prediction methods.

S.2 The experimental results show that sparse graphs contain strong caps on optimal performance, which will induce more consideration of other information beyond the topological structure for real sparse graphs during model design.

S.3 The experimental analysis is detailed and clear, and the results are fully explained.

Weaknesses:

W.1 The current mainstream link prediction methods are not only based on graph topology, the relations and potential impact on these topology-beyond methods haven't been discussed in this paper.

W.2 Only one topology-based link-prediction method is evaluated for Sec.6.2, the conclusions of Sec.6.2 are less convincing.

W.3 The structure of the article is a little incoherent and not easy to understand. The motivation of proposed method is not clearly explained.

---

> ### Author Response · Authors · 2023-04-13
> **Reply to Review 3**
>
> # Review 3
>
> *Each of these three reviews was nuanced and helpful. Thank you!*
>
> ## Summary Of Contributions
>
> This paper proposes hard upper bounds of link prediction performance based on topology, which brings new insight on analyzing the utilization of graph structure information for link prediction. The authors conduct experiments on a variety of real-world graphs to calculate how maximum link prediction scores vary with the amount of information based on proposed optimal prediction performance measurements. The experimental results demonstrate that link prediction methods on sparse graphs are inherently limited and require information besides the graph topology.
>
> ## Strengths And Weaknesses
>
> ### Strengths
>
> S.1 This paper provides a new approach to analyze topology-based link prediction methods and the upper bounds will help the evaluation of link prediction methods.
>
> S.2 The experimental results show that sparse graphs contain strong caps on optimal performance, which will induce more consideration of other information beyond the topological structure for real sparse graphs during model design.
>
> S.3 The experimental analysis is detailed and clear, and the results are fully explained.
>
> ### Weaknesses
>
> W.1 The current mainstream link prediction methods are not only based on graph topology, the relations and potential impact on these topology-beyond methods haven't been discussed in this paper.
>
> *For what it’s worth, we do not think of this as a weakness of the paper for the following reason: Our whole goal was to explore the limits of what can be inferred from the structure. Thus while we know that almost all modern link prediction methods make use non-structural data, that is almost beside the point of this study.*
>
> W.2 Only one topology-based link-prediction method is evaluated for Sec.6.2, the conclusions of Sec.6.2 are less convincing.
>
> *[We discuss how we can make section 6.2 more convincing in our response to Review 2’s last 2 points.]*
>
> *We admit that it would be ideal to compare to more link prediction models. However, we also would like to point out that link prediction performance tends to be similar across GNN’s (within a few percent – whereas the amount by which GCNAE’s reported score was above the limit was 15%), and our goal in the paper was not to analyze GCNAE per-say but rather to show that:*
>
> *“Firstly, a metric (e.g. AP) may have different meanings depending on how it is used, and our methodology may be able to help retroactively determine which approach was used if the original paper does not specify.*
>
> *Secondly, and perhaps of greater interest, state of the art link prediction systems using the topology of a network do not reach the topology-based upper limit on performance....”*
>
> *We believe that both of these claims will be well-established through our updated analysis of GCNAE (see Reply to Review 2 -- Part 2).*
>
> *However, we agree that we should at minimum add a note to the paper establishing the similarity between GCNAE and other models’ performance.*
>
> W.3 The structure of the article is a little incoherent and not easy to understand. The motivation of proposed method is not clearly explained.
>
> *We see that the need for improved clarity is mentioned in all 3 reviews and intend to make a serious effort to improve these sections.*
>
> ## Requested Changes
>
> There are several suggestions
>
> 1. The relations and potential impact on topology-beyond link prediction methods should be discussed in this paper.
>
> 2. More topology-based methods need to be considered to provide more convincing conclusions.
>
> 3. The article structure and writing need to be improved so that readers can understand the contribution of the article more easily.
>
> *[See the above for our thoughts and intentions to address these concerns.]*

---

### Review · Reviewer_sTw9 · 2023-03-10

**Summary Of Contributions:**

The authors study upper bounds for common metrics like ROC and AUPR which are used for the task of link prediction within the class of methods that are invariant to permutations of the labels of nodes.


**Audience:**

Yes

**Broader Impact Concerns:**

No ethical concerns.

**Claims And Evidence:**

No

**Requested Changes:**

See above.

**Strengths And Weaknesses:**

+ I found interesting the idea of studying the best possible result that can be achieved by methods that are invariant to permutations of the nodes. Especially on datasets that are still frequently used in the literature, e.g., Cora and CiteSeer.

- I have some issues with the writing of the paper. Especially the Introduction and Section 4. The authors are trying to measure best possible performance among methods that are invariant to node permutations in the graph. So it seems that the symmetry that they are interested at is node permutations. This is a very well studied subject in GNN literature and others (see the work of Haggai Maron and citations to him). So I don't understand why the authors don't mention this clearly in the introduction and clearly re-frame their paper within this context? I would recommend to the authors to study the GNN literature more and place their work appropriately.

- Section 4 is very hard to read. So I simply understood the high level idea and skipped the details. The authors need to provide further explanations about equations 1, 2 and 3, otherwise, these are very difficult to interpret. These are the most important and only mathematical results in this paper.

- What's an anonymized version of a graph? It would be good to define any terminology that is used.
- The word "maximally" is used but no concrete definition of what maximally means here is given. I get the idea that the authors utilize ground truth information for a specific labelling of the graph, but why is this "maximal"? Usually this kind of definition implies that the given information are provably better that any other given information, but is it the case here? It seems more appropriate to simply calls this "informed link prediction".
- Typo "maximially"
- It would have been very helpful if the "Original graph" in Fig 1 is plotted using the same 2D coordinates as in the other parts of the figure.
- The definition of k-hop neighbours is very difficult to understand. It seems unnecessarily complex.
- What's a "canonical representation"?

- Lastly, one the most important weakness is the analysis and discussion compared to GCNAE. First, I observed that the authors make a lot of hypothesis about what could have been implemented in the GCNAE paper. However, with just a quick search I found the code for the paper https://github.com/tkipf/gae that includes the data splitting that was used and a lot of programming details. So I don't understand why the authors need to make all these assumptions.
- Secondly, GCNAE without node features is not permutation equivariant. So I don't think it makes sense to check its performance in this setting where node permutation symmetries are important. The authors do mention this in their "Caveats" Section, but it is stated as hypothesis. As I mentioned earlier, there is a lot of work on this subject and I would recommend the authors to read the literature on GNNs for invariant and equivariant GNNs.

---

> ### Author Response · Authors · 2023-04-13
> **Reply to Review 2 -- Part 1**
>
> # Review 2
>
> *Each of these three reviews was nuanced and helpful. Thank you!*
>
> ## Summary Of Contributions
>
> The authors study upper bounds for common metrics like ROC and AUPR which are used for the task of link prediction within the class of methods that are invariant to permutations of the labels of nodes.
>
> ## Strengths And Weaknesses
> • I found interesting the idea of studying the best possible result that can be achieved by methods that are invariant to permutations of the nodes. Especially on datasets that are still frequently used in the literature, e.g., Cora and CiteSeer.
>
> • I have some issues with the writing of the paper. Especially the Introduction and Section 4. The authors are trying to measure best possible performance among methods that are invariant to node permutations in the graph. So it seems that the symmetry that they are interested at is node permutations. This is a very well studied subject in GNN literature and others (see the work of Haggai Maron and citations to him). So I don't understand why the authors don't mention this clearly in the introduction and clearly re-frame their paper within this context? I would recommend to the authors to study the GNN literature more and place their work appropriately.
>
> *We will work to better frame our paper vis-a-vis the GNN literature. However, as we comment about above in response to Review 1:*
>
> *“It [GNN research] is related, but not directly related as far as we know. For example, much of the research on the limits of GNNs [e.g. Haggai Maron] have to do with the correspondence between a GNN architecture and the power of a k-dimensional Weisfeiler Lehman (WL) algorithm. The family of WL algorithms are used to determine information about the graph’s symmetries up to isomorphism. Thus the GNN research considers how much of the available structural symmetry data a GNN architecture can take advantage of. Our research considers a different question: If all the symmetry data is available, what limits remain?”*
>
> *Perhaps more importantly, we should also flag several factors that drastically mitigate the relevance of the fact that our proof is about node permutation-invariant algorithms.*
>
> *A. There are only two ways that an algorithm which is not permutation-invariant can score better than an algorithm that is permutation-invariant:*
>
> *A1. The input graph’s node ordering was based on some property of the solution graph.*
>
> *A2. The input ordering had no significance but the algorithm gets lucky arbitrarily due to the input (and would have performed worse given a different arbitrary input node ordering).*
>
> *Concerning A1: Nobody should want algorithms to make use of the kind of data in A1 because that data is not available in real-world link prediction settings (e.g. a sales website cannot use a node ordering based on what products you will buy).*
>
> *Concerning A2: The possibility that an algorithm could get lucky is not a meaningful counter-example to a performance limit.*
>
> *Thus neither reason that an algorithm which is non-permutation-invariant could hypothetically surpass our bound is a meaningful reason. Consequently our bound is effectively an upper-bound for all link prediction algorithms – not just permutation-invariant ones.*
>
>
> • Section 4 is very hard to read. So I simply understood the high level idea and skipped the details. The authors need to provide further explanations about equations 1, 2 and 3, otherwise, these are very difficult to interpret. These are the most important and only mathematical results in this paper.
>
> *We see that the need for improved clarity is mentioned in all 3 reviews and intend to make a serious effort to improve these sections.*

---

> ### Author Response · Authors · 2023-04-13
> **Reply to Review 2 -- Part 2**
>
> • What's an anonymized version of a graph? It would be good to define any terminology that is used.
>
> *Good point. We will make this clearer in the paper. Given a graph G, we think of an “anonymized” version as being a graph H that is isomorphic to G where no exact isomorphism between G and H is given. Thus the structure of the graph is preserved but the identity of a given node is not known (“anonymous”) except insofar as it is entailed by the structure.*
>
> • The word "maximally" is used but no concrete definition of what maximally means here is given. I get the idea that the authors utilize ground truth information for a specific labeling of the graph, but why is this "maximal"? Usually this kind of definition implies that the given information are provably better that any other given information, but is it the case here? It seems more appropriate to simply calls this "informed link prediction".
>
> *Good question. We called it “maximally informed” link prediction because the algorithm is given all possible structural information entailed by the problem alone (i.e. the solution graph itself showing which edges were removed). Any other source of information is either not strictly structural (e.g. text data related to nodes) or not entailed by the task alone (e.g. a node ordering based on an algorithm).*
>
> *Thus, in terms of structural information, this is provably better simply because it is all the possible information. However, we think you are right that perhaps “maximally informed” is misleading because there are other kinds of non-structural data a link predictor could be given.*
>
> *We could instead pick a name like “maximally informed structure-only link prediction,” but that’s a mouthful. We will brainstorm to try to come up with a better name.*
>
> • Typo "maximially"
>
> • It would have been very helpful if the "Original graph" in Fig 1 is plotted using the same 2D coordinates as in the other parts of the figure.
>
> *We will make this change.*
>
> • The definition of k-hop neighbours is very difficult to understand. It seems unnecessarily complex.
>
> *Acknowledged. We will work to simplify this via some combination of:*
>
> *1. Reducing the amount of formalism*
>
> *2. Talking about walks of length k instead of recursive neighbor sets and defining induced subgraphs in their own preliminary section so that the notation is simpler*
>
> *3. Something you suggest*
>
> • What's a "canonical representation"?
>
> *Good question. We will add an explanation to the paper. A “canonical representation” of a graph (or of a subgraph) is a representation of the graph that is produced in a way that is invariant to the original node ordering of the graph. This concept is used by practical graph isomorphism algorithms (e.g. Nauty and Traces); they work by first converting two graphs G1 and G2 to canonical forms C1 and C2 respectively, then perform the trivial check to see if C1 and C2 are identical.*
>
> • Lastly, one the most important weakness is the analysis and discussion compared to GCNAE. First, I observed that the authors make a lot of hypothesis about what could have been implemented in the GCNAE paper. However, with just a quick search I found the code for the paper https://github.com/tkipf/gae (https://github.com/tkipf/gae) that includes the data splitting that was used and a lot of programming details. So I don't understand why the authors need to make all these assumptions.
>
> *True. We were probably too lazy in this regard. We are now willing to extract the answers from their code.*
>
> • Secondly, GCNAE without node features is not permutation equivariant. So I don't think it makes sense to check its performance in this setting where node permutation symmetries are important. The authors do mention this in their "Caveats" Section, but it is stated as hypothesis. As I mentioned earlier, there is a lot of work on this subject and I would recommend the authors to read the literature on GNNs for invariant and equivariant GNNs.
>
> *There is a fairly straightforward way for us to address this concern. We can make a modified version of GCNAE that first randomly permutes the input node ordering and call it GCNAE'.*
>
> *We strongly suspect that GCNAE' will report similar scores to the scores reported in the original paper. If it reports similar scores, this would show that the amount reported over our performance limit is not due to input ordering but rather due to down-sampling negative edges. If GCNAE' does not report similar scores, the new numbers will still be interesting and should be considered the most relevant scores due to the reasons we discuss above concerning A1.*
>
> *Furthermore, actually running GCNAE and GCNAE' will enable us to see what its AP score actually is when calculated correctly.*

---

### Review · Reviewer_TEcm · 2023-04-10

**Summary Of Contributions:**

This paper considers what inherently limits predictability in link prediction tasks, questioning whether graph automorphisms and related properties yield inherent limitations in predictability. For real-world graphs, they calculated upper bounds for whether graph topology alone can make link predictions. One finding was that predictions for datasets whose graphs are sparse are inherently limited and require information in addition to graph topology.

**Audience:**

No

**Broader Impact Concerns:**

I don't see a problem on impact concerns.

**Claims And Evidence:**

Yes

**Requested Changes:**


1.The structure of the introductory section should be improved:
For example:

-It is not clear until Section 3 that we see what the maximally informed link prediction task is mathematically, but it would be helpful if it is explained earlier.

-The link prediction problem has a lot of existing research and history, but I feel that link prediction itself is not well explained.  It would be great if the authors give an explanation of link prediction itself and an introduction of existing studies and representative methods would be helpful. If there is anything that interferes with the flow of the discussion, create a Related Work.

 -A paragraph that concisely summarizes the contributions of the paper is needed. For example, how the proposed methodology gives insights about topology-based link prediction should be briefly described.

-It would be easier to read if you include a reference to the structure of the paper.

2. Experiments

Although the paper found that sparseness is characteristic to differentiate on inherent limits, it would be interesting to explore other notions of graphs such as clustering coefficient or average degree might be related. I suspect these notions are not making much difference, but even so, the authors can provide such basic topological information and add them to the Table for graph data.


**Strengths And Weaknesses:**

Strengths

-It is fascinating to discuss the marginality of the task of link prediction, focusing only on the topological information.

-They experimentally show how the AUPR Score changes with the amount of information in the variety of graphs.

-Furthermore, they experimentally showed that ROC and AP are calculated using Graph Convolutional Neural Network Auto-Encoder (GCNAE), and that a gap between the ROC and AP is observed.


Weaknesses

-This paper focuses on topologically-based link prediction, whereas there are many different problem settings for link prediction that are more common to use features. How to deal with feature information is sometimes the key to learning, and several non-topological features improve the accuracy of link prediction substantially as shown in existing work (e.g. Al Hasan, Mohammad, et al. "Link prediction using supervised learning." SDM06: workshop on link analysis, counter-terrorism and security. Vol. 30. 2006.)

On the other hand, in this paper, the motivation should be more clear for discussing learning limitedness by restricting the discussion to topological information only.

--Is this paper completely unrelated to the research stream with Graph Neural Networks? I would like to confirm that similar research questions have not been addressed in GNN literature.

---

> ### Author Response · Authors · 2023-04-13
> **Reply to Review #1**
>
> # Review 1
>
> *Each of these three reviews was nuanced and helpful. Thank you!*
>
> # Summary Of Contributions
>
> This paper considers what inherently limits predictability in link prediction tasks, questioning whether graph automorphisms and related properties yield inherent limitations in predictability. For real-world graphs, they calculated upper bounds for whether graph topology alone can make link predictions. One finding was that predictions for datasets whose graphs are sparse are inherently limited and require information in addition to graph topology.
>
> # Strengths And Weaknesses
>
> ## Strengths
>
> -It is fascinating to discuss the marginality of the task of link prediction, focusing only on the topological information.
>
> -They experimentally show how the AUPR Score changes with the amount of information in the variety of graphs.
>
> -Furthermore, they experimentally showed that ROC and AP are calculated using Graph Convolutional Neural Network Auto-Encoder (GCNAE), and that a gap between the ROC and AP is observed.
>
> ## Weaknesses
>
> -This paper focuses on topologically-based link prediction, whereas there are many different problem settings for link prediction that are more common to use features. How to deal with feature information is sometimes the key to learning, and several non-topological features improve the accuracy of link prediction substantially as shown in existing work (e.g. Al Hasan, Mohammad, et al. "Link prediction using supervised learning." SDM06: workshop on link analysis, counter-terrorism and security. Vol. 30. 2006.)
> On the other hand, in this paper, the motivation should be more clear for discussing learning limitedness by restricting the discussion to topological information only.
>
> *For what it’s worth, we do not think of this as a weakness of the paper for the following reason: Our whole goal was to explore the limits of what can be inferred from the structure. Thus while we know that almost all modern link prediction methods make use non-structural data, that is almost beside the point of this study.*
>
> --Is this paper completely unrelated to the research stream with Graph Neural Networks? I would like to confirm that similar research questions have not been addressed in GNN literature.
>
> *It is related, but not directly related as far as we know. For example, much of the research on the limits of GNNs have to do with the correspondence between a GNN architecture and the power of a k-dimensional Weisfeiler Lehman (WL) algorithm. The family of WL algorithms are used to determine information about the graph’s symmetries up to isomorphism. Thus the GNN research considers how much of the available structural symmetry data a GNN architecture can take advantage of. Our research considers a different question: If all the symmetry data is available, what limits remain?*
>
>
> # Requested Changes
>
> 1.The structure of the introductory section should be improved: For example:
>
> -It is not clear until Section 3 that we see what the maximally informed link prediction task is mathematically, but it would be helpful if it is explained earlier.
>
> *We see that the need for improved clarity is mentioned in all 3 reviews and intend to make a serious effort to improve these sections.*
>
> -The link prediction problem has a lot of existing research and history, but I feel that link prediction itself is not well explained. It would be great if the authors give an explanation of link prediction itself and an introduction of existing studies and representative methods would be helpful. If there is anything that interferes with the flow of the discussion, create a Related Work.
>
> *Precisely because there is so much research on the subject, we thought that the link prediction task is so well-known that it does not need to be re-explained or re-introduced. However, we are willing to adjust this aspect of the paper and add more exposition on the subject.*
>
> -A paragraph that concisely summarizes the contributions of the paper is needed. For example, how the proposed methodology gives insights about topology-based link prediction should be briefly described.
> -It would be easier to read if you include a reference to the structure of the paper.
>
> *Thanks for the feedback. We are happy to add these paragraphs.*
>
>
> 2. Experiments
>
> Although the paper found that sparseness is characteristic to differentiate on inherent limits, it would be interesting to explore other notions of graphs such as clustering coefficient or average degree might be related. I suspect these notions are not making much difference, but even so, the authors can provide such basic topological information and add them to the Table for graph data.
>
> *Thanks for this suggestion. When we revise the paper, we will add this sort of information.*

---

### Decision · Action_Editors · 2023-05-17

**Recommendation:** Accept with minor revision

**Comment:**

In agreement with the reviewers assessment, I believe that the paper makes some valuable contribution, yet needs some polishing and the inclusion of additional discussions before being accepted, according to the suggestions made by the reviewers, and as discussed in the authors' replies. I copy below the exchanges where additional discussion has been requested and the authors' constructive replies and suggestions as to how they intend to reply to these requests. I essentially ask the authors to implement the suggestions they laid out, with a particular emphasis on improving the paper's readability.

Reviewer 1
1.The structure of the introductory section should be improved: For example:

-It is not clear until Section 3 that we see what the maximally informed link prediction task is mathematically, but it would be helpful if it is explained earlier.

We see that the need for improved clarity is mentioned in all 3 reviews and intend to make a serious effort to improve these sections.

-The link prediction problem has a lot of existing research and history, but I feel that link prediction itself is not well explained. It would be great if the authors give an explanation of link prediction itself and an introduction of existing studies and representative methods would be helpful. If there is anything that interferes with the flow of the discussion, create a Related Work.

Precisely because there is so much research on the subject, we thought that the link prediction task is so well-known that it does not need to be re-explained or re-introduced. However, we are willing to adjust this aspect of the paper and add more exposition on the subject.

-A paragraph that concisely summarizes the contributions of the paper is needed. For example, how the proposed methodology gives insights about topology-based link prediction should be briefly described. -It would be easier to read if you include a reference to the structure of the paper.

Thanks for the feedback. We are happy to add these paragraphs.

Experiments
Although the paper found that sparseness is characteristic to differentiate on inherent limits, it would be interesting to explore other notions of graphs such as clustering coefficient or average degree might be related. I suspect these notions are not making much difference, but even so, the authors can provide such basic topological information and add them to the Table for graph data.

Thanks for this suggestion. When we revise the paper, we will add this sort of information.

Reviewer 2:
• I found interesting the idea of studying the best possible result that can be achieved by methods that are invariant to permutations of the nodes. Especially on datasets that are still frequently used in the literature, e.g., Cora and CiteSeer.

• I have some issues with the writing of the paper. Especially the Introduction and Section 4. The authors are trying to measure best possible performance among methods that are invariant to node permutations in the graph. So it seems that the symmetry that they are interested at is node permutations. This is a very well studied subject in GNN literature and others (see the work of Haggai Maron and citations to him). So I don't understand why the authors don't mention this clearly in the introduction and clearly re-frame their paper within this context? I would recommend to the authors to study the GNN literature more and place their work appropriately.

We will work to better frame our paper vis-a-vis the GNN literature. However, as we comment about above in response to Review 1:

“It [GNN research] is related, but not directly related as far as we know. For example, much of the research on the limits of GNNs [e.g. Haggai Maron] have to do with the correspondence between a GNN architecture and the power of a k-dimensional Weisfeiler Lehman (WL) algorithm. The family of WL algorithms are used to determine information about the graph’s symmetries up to isomorphism. Thus the GNN research considers how much of the available structural symmetry data a GNN architecture can take advantage of. Our research considers a different question: If all the symmetry data is available, what limits remain?”

Perhaps more importantly, we should also flag several factors that drastically mitigate the relevance of the fact that our proof is about node permutation-invariant algorithms.

A. There are only two ways that an algorithm which is not permutation-invariant can score better than an algorithm that is permutation-invariant:

A1. The input graph’s node ordering was based on some property of the solution graph.

A2. The input ordering had no significance but the algorithm gets lucky arbitrarily due to the input (and would have performed worse given a different arbitrary input node ordering).

Concerning A1: Nobody should want algorithms to make use of the kind of data in A1 because that data is not available in real-world link prediction settings (e.g. a sales website cannot use a node ordering based on what products you will buy).

Concerning A2: The possibility that an algorithm could get lucky is not a meaningful counter-example to a performance limit.

Thus neither reason that an algorithm which is non-permutation-invariant could hypothetically surpass our bound is a meaningful reason. Consequently our bound is effectively an upper-bound for all link prediction algorithms – not just permutation-invariant ones.

• Section 4 is very hard to read. So I simply understood the high level idea and skipped the details. The authors need to provide further explanations about equations 1, 2 and 3, otherwise, these are very difficult to interpret. These are the most important and only mathematical results in this paper.

We see that the need for improved clarity is mentioned in all 3 reviews and intend to make a serious effort to improve these sections.
• What's an anonymized version of a graph? It would be good to define any terminology that is used.

Good point. We will make this clearer in the paper. Given a graph G, we think of an “anonymized” version as being a graph H that is isomorphic to G where no exact isomorphism between G and H is given. Thus the structure of the graph is preserved but the identity of a given node is not known (“anonymous”) except insofar as it is entailed by the structure.

• The word "maximally" is used but no concrete definition of what maximally means here is given. I get the idea that the authors utilize ground truth information for a specific labeling of the graph, but why is this "maximal"? Usually this kind of definition implies that the given information are provably better that any other given information, but is it the case here? It seems more appropriate to simply calls this "informed link prediction".

Good question. We called it “maximally informed” link prediction because the algorithm is given all possible structural information entailed by the problem alone (i.e. the solution graph itself showing which edges were removed). Any other source of information is either not strictly structural (e.g. text data related to nodes) or not entailed by the task alone (e.g. a node ordering based on an algorithm).

Thus, in terms of structural information, this is provably better simply because it is all the possible information. However, we think you are right that perhaps “maximally informed” is misleading because there are other kinds of non-structural data a link predictor could be given.

We could instead pick a name like “maximally informed structure-only link prediction,” but that’s a mouthful. We will brainstorm to try to come up with a better name.

• Typo "maximially"

• It would have been very helpful if the "Original graph" in Fig 1 is plotted using the same 2D coordinates as in the other parts of the figure.

We will make this change.

• The definition of k-hop neighbours is very difficult to understand. It seems unnecessarily complex.

Acknowledged. We will work to simplify this via some combination of:

1. Reducing the amount of formalism

2. Talking about walks of length k instead of recursive neighbor sets and defining induced subgraphs in their own preliminary section so that the notation is simpler

3. Something you suggest

• What's a "canonical representation"?

Good question. We will add an explanation to the paper. A “canonical representation” of a graph (or of a subgraph) is a representation of the graph that is produced in a way that is invariant to the original node ordering of the graph. This concept is used by practical graph isomorphism algorithms (e.g. Nauty and Traces); they work by first converting two graphs G1 and G2 to canonical forms C1 and C2 respectively, then perform the trivial check to see if C1 and C2 are identical.

• Lastly, one the most important weakness is the analysis and discussion compared to GCNAE. First, I observed that the authors make a lot of hypothesis about what could have been implemented in the GCNAE paper. However, with just a quick search I found the code for the paper https://github.com/tkipf/gae (https://github.com/tkipf/gae) that includes the data splitting that was used and a lot of programming details. So I don't understand why the authors need to make all these assumptions.

True. We were probably too lazy in this regard. We are now willing to extract the answers from their code.

• Secondly, GCNAE without node features is not permutation equivariant. So I don't think it makes sense to check its performance in this setting where node permutation symmetries are important. The authors do mention this in their "Caveats" Section, but it is stated as hypothesis. As I mentioned earlier, there is a lot of work on this subject and I would recommend the authors to read the literature on GNNs for invariant and equivariant GNNs.

There is a fairly straightforward way for us to address this concern. We can make a modified version of GCNAE that first randomly permutes the input node ordering and call it GCNAE'.

We strongly suspect that GCNAE' will report similar scores to the scores reported in the original paper. If it reports similar scores, this would show that the amount reported over our performance limit is not due to input ordering but rather due to down-sampling negative edges. If GCNAE' does not report similar scores, the new numbers will still be interesting and should be considered the most relevant scores due to the reasons we discuss above concerning A1.

Furthermore, actually running GCNAE and GCNAE' will enable us to see what its AP score actually is when calculated correctly.
Reviewer 3:
W.1 The current mainstream link prediction methods are not only based on graph topology, the relations and potential impact on these topology-beyond methods haven't been discussed in this paper.

For what it’s worth, we do not think of this as a weakness of the paper for the following reason: Our whole goal was to explore the limits of what can be inferred from the structure. Thus while we know that almost all modern link prediction methods make use non-structural data, that is almost beside the point of this study.

W.2 Only one topology-based link-prediction method is evaluated for Sec.6.2, the conclusions of Sec.6.2 are less convincing.

[We discuss how we can make section 6.2 more convincing in our response to Review 2’s last 2 points.]

We admit that it would be ideal to compare to more link prediction models. However, we also would like to point out that link prediction performance tends to be similar across GNN’s (within a few percent – whereas the amount by which GCNAE’s reported score was above the limit was 15%), and our goal in the paper was not to analyze GCNAE per-say but rather to show that:

“Firstly, a metric (e.g. AP) may have different meanings depending on how it is used, and our methodology may be able to help retroactively determine which approach was used if the original paper does not specify.

Secondly, and perhaps of greater interest, state of the art link prediction systems using the topology of a network do not reach the topology-based upper limit on performance....”

We believe that both of these claims will be well-established through our updated analysis of GCNAE (see Reply to Review 2 -- Part 2).

However, we agree that we should at minimum add a note to the paper establishing the similarity between GCNAE and other models’ performance.

W.3 The structure of the article is a little incoherent and not easy to understand. The motivation of proposed method is not clearly explained.

We see that the need for improved clarity is mentioned in all 3 reviews and intend to make a serious effort to improve these sections.

Requested Changes
There are several suggestions

The relations and potential impact on topology-beyond link prediction methods should be discussed in this paper.

More topology-based methods need to be considered to provide more convincing conclusions.

The article structure and writing need to be improved so that readers can understand the contribution of the article more easily.

[See the above for our thoughts and intentions to address these concerns.]


**Audience:**

yes

**Claims And Evidence:**

yes